

# Prediction of PM$_{2.5}$ concentration based on the CEEMDAN-RLMD-BiLSTM-LEC model

Qiao Guo, Haoyu Zhang, Yuhao Zhang and Xuchu Jiang

School of Statistics and Mathematics, Zhongnan University of Economics and Law,
Wuhan, China

## ABSTRACT

Air quality has emerged as a critical concern in recent years, with the concentration of PM$_{2.5}$ recognized as a vital index for assessing it. The accuracy of predicting PM$_{2.5}$ concentrations holds significant value for effective air quality monitoring and management. In response to this, a combined model comprising CEEMDAN-RLMD-BiLSTM-LEC has been introduced, analyzed, and compared against various other models. The combined decomposition method effectively underlines the fundamental characteristics of the data compared to individual decomposition techniques. Additionally, local error correction (LEC) efficiently addresses the issue of prediction errors induced by excessive disturbances. The empirical results of nine steps indicate that the combined CEEMDAN-RLMD-BiLSTM-LEC model outperforms single prediction models such as RLMD and CEEMDAN, reducing MAE, RMSE, and SAMPE by 36.16%, 28.63%, 45.27% and 16.31%, 6.15%, 37.76%, respectively. Moreover, the inclusion of LEC in the model further diminishes MAE, RMSE, and SMAPE by 20.69%, 7.15%, and 44.65%, respectively, exhibiting commendable performance in generalization experiments. These findings demonstrate that the combined CEEMDAN-RLMD-BiLSTM-LEC model offers high predictive accuracy and robustness, effectively handling noisy data predictions and severe local variations. With its wide applicability, this model emerges as a potent tool for addressing various related challenges in the field.

# INTRODUCTION

PM$_{2.5}$ is the main source of air pollution, which poses significant risks to human lungs and safety. Therefore, accurately forecasting the concentration of pollutants such as PM$_{2.5}$ can efficiently prompt people to take preventive measures in advance.

Several studies classify PM$_{2.5}$ prediction models into three categories: numerical, statistical, and machine learning models. Numerical models require detailed emissions data and comprehensive knowledge of pollutant mechanisms for model configuration (*Yumimoto & Uno, 2006*). However, accurately representing pollutant emissions on different spatial and temporal scales remains challenging. *Xu et al. (2008)* made advances in estimating emissions, but biases persist, including weather system prediction bias,

Corresponding author
Xuchu Jiang,
xuchujiang@zuel.edu.cn

real-time emissions portrayal limitations, and errors in numerical model parameterization. These challenges lead to significant discrepancies in numerical model predictions for specific regions.

Statistical models provide a more convenient option. *Van Donkelaar et al. (2010)* discovered a linear correlation between $PM_{2.5}$, a significant component of aerosols, and the multiphase system of gas, liquid, and solid particles suspended in the atmosphere (AOD). Consequently, they proposed using AOD to predict $PM_{2.5}$ concentration. Additionally, *Ma et al. (2014)* utilized the geographically weighted regression model for $PM_{2.5}$ prediction, with a coefficient of determination of 0.64. *Zhao et al. (2018)* used aerosol optical depth data, meteorological factors monitored at ground level (wind speed, temperature, and relative humidity), and other gaseous pollutants ($SO_2$, $NO_2$, CO, and $O_3$) to predict $PM_{2.5}$, resulting in a coefficient of determination of 0.76. To some extent, regression models heavily rely on the data sources and accuracy of each variable. However, tracing the sources of $PM_{2.5}$ is difficult due to substantial variations in concentrations across different regions and time periods (*Engel-Cox et al., 2013*). Moreover, measurement accuracy can be compromised by the high measurement noise associated with AOD (*Munchak et al., 2013*).

In recent years, machine learning and deep learning models have gained recognition for their superior performance in nonlinear data fitting and prediction, *Zamani Joharestani et al. (2019)* used random forests, gradient boosting, and machine learning to predict $PM_{2.5}$ concentrations in the Tehran metropolitan area. They found the best model performance using the XGBoost with 23 onsite measured $PM_{2.5}$ and geographic data features, reaching an $R^2 = 0.81$. *Yang et al. (2018)* proposed a spatiotemporal support vector regression model and constructed a Gaussian vector weight function considering factors such as distance and wind direction. *Yu et al. (2022)* built a $PM_{2.5}$ prediction model combining fast Fourier transform and LSTM neural networks. *Wang et al. (2013)* used the BP artificial neural network for $PM_{2.5}$ prediction and compared it to ordinary kriging interpolation, highlighting the superiority of the machine learning model. *Zhang et al. (2020)* applied principal component analysis for data dimensionality reduction, combining this with a BP neural network to analyze the impact of various seasons and atmospheric factors on $PM_{2.5}$ concentration.

Significant signal decomposition methods such as empirical mode decomposition (EMD) have successfully handled nonstationary signals to some degree, leading to their application across various domains. *Chen et al. (2016)* used Spearman-Rank analysis and complete ensemble empirical mode decomposition combined with adaptive noise to study the spatiotemporal distribution characteristics of $PM_{2.5}$ and the influence of meteorological factors on $PM_{2.5}$ in Nanjing.

In other studies, the bidirectional long short-term memory model (BiLSTM) has been employed for $PM_{2.5}$ prediction. *Prihatno et al. (2021)* developed a single density layer BiLSTM for indoor $PM_{2.5}$ prediction, which showed low errors and applicability to practical research. *Wang et al. (2022)* used $PM_{2.5}$ data from four cities and proposed a hybrid multiscale learning framework based on robust local mean decomposition (RLMD) and a moving window ensemble strategy. *Ban & Shen (2022)* suggested that it is
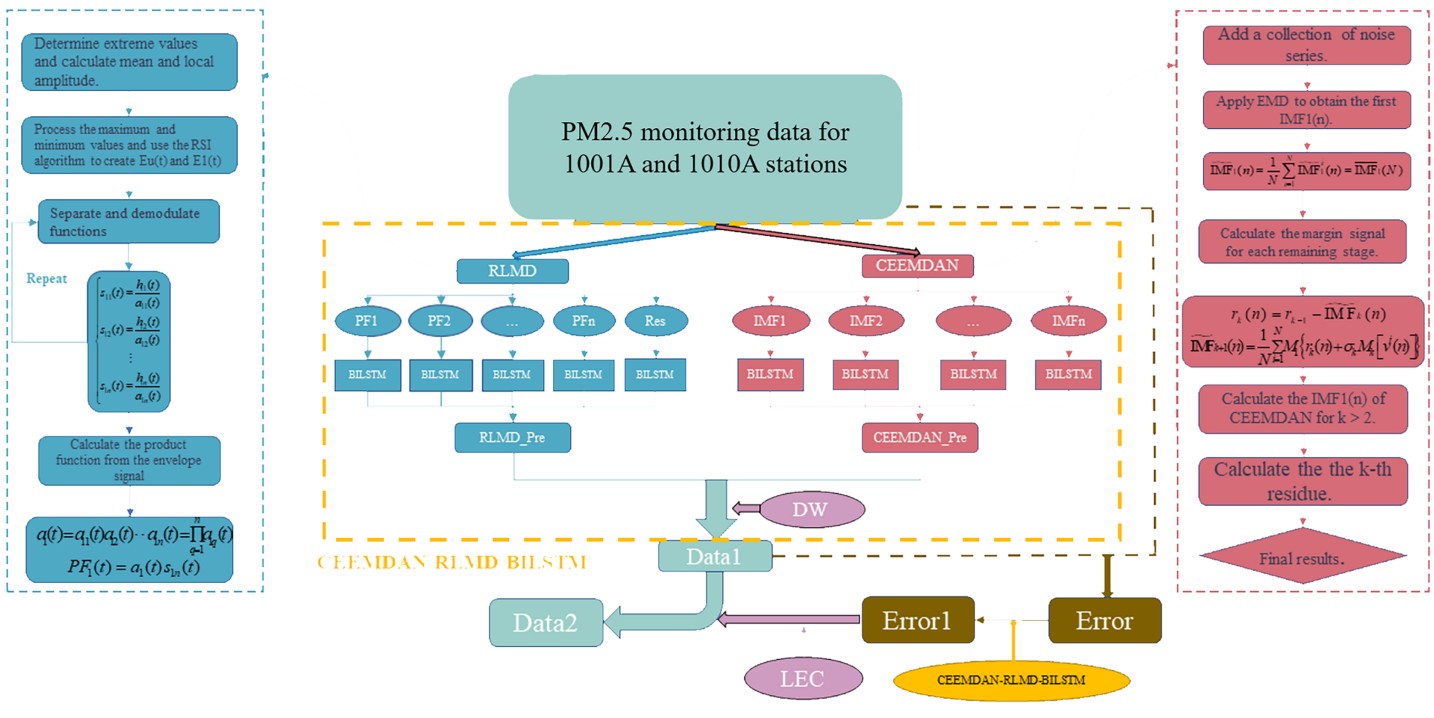

**Figure 1 Basic flow of this article.**

challenging to conduct PM$_{2.5}$ prediction tasks using only one model. Therefore, they compared complete ensemble empirical mode decomposition with adaptive noise (CEEMDAN) with eight different models and found it performed best.

From the above research, it can be concluded that for the prediction of PM$_{2.5}$ concentration, numerical models and statistical models require high data accuracy and sensitivity, resulting in relatively lower prediction accuracy in practice. On the other hand, single models based on machine learning and deep learning exhibit limited feature capturing capabilities, and common machine learning models have not demonstrated superiority in practical research. To improve prediction accuracy and minimize potential errors, this study combines CEEMDAN and RLMD to optimize measurement stability and proposes the CEEMDAN-RLMD-BiLSTM-LEC model for predicting PM$_{2.5}$ sequence data detected at stations 1001A and 1010A in Beijing (http://www.bjmemc.com.cn/). The flowchart of this process is illustrated in Fig. 1.

## METHODS

### CEEMDAN-RLMD-BiLSTM

#### CEEMDAN

Due to the "modal overlap" caused by the EMD algorithm and the noise residual caused by EEMD, this article introduces the CEEMDAN algorithm (*Torres et al., 2011*), which overcomes the defects of EEMD decomposition in terms of loss of completeness and modal aliasing by adaptively adding white noise.

### RLMD

LMD is an improved algorithm based on EMD, which is a decomposition of the signal into a series of product function components (PF) (*Smith, 2005*). When each PF component is separated from the original signal, the residual component of the signal $u_k(t)$ is obtained, which indicates that the original signal can be considered the sum of the residual component and all PF components, *i.e.*, as Eq. (1).

$$x(t) = \sum_{p=1}^{k} \mathrm{PF}_p(t) + u_k(t) \tag{1}$$

The main influencing factors of LMD are mainly three points, which are boundary conditions, envelope estimation, and stopping criterion of screening. The specific optimization steps of the robust mean algorithm (RLMD) are as follows: boundary conditions: determine the symmetry points of the left and right ends of the signal using the mirror expansion algorithm; envelope estimation: obtain the optimal subset according to statistical theory; screening stopping principle: minimize the error function, and then RLMD can automatically determine the fixed subset size of the moving average algorithm and the optimal number of screening iterations in the screening process, so it becomes a time-frequency effective tool for analysis (*Liu et al., 2017*).

### BiLSTM

Compared with the traditional unidirectional LSTM neural network, adding a reverse LSTM layer on top of the original one can make it have a two-way propagation loop structure (*Zhao et al., 2017*). The BiLSTM allows the past and future states of the implicit layer to be passed and fed back through a bidirectional network.

### Dynamic weighting method

CEEMDAN effectively solves the modal blending problem and endpoint effect of the EMD algorithm and extracts the feature information of the original signal more fully, but there are problems of noise redundancy and spurious components, while RLMD can effectively solve the problem of spurious components among them and achieve optimization for CEEMDAN. Therefore, this article organically combines the prediction results of the two by the dynamic weighting method. *pred(t)* of the CEEMDAN-RLMD-BiLSTM prediction results are as Eq. (2).

$$pred(t) = w_1 pred_1(t) + w_2 pred_2(t) \tag{2}$$

where $pred_1(t)$ is the CEEMDAN prediction result, $pred_2(t)$ is the RLMD prediction result, and $w_1$ and $w_2$ are the weights. The weights are calculated by taking the initial $w_1 = 1$ and $w_2 = 0$, varying them in steps of 0.01 to find the corresponding *pred(t)*, and using the root mean square error (RMSE) as the criterion. The weight corresponding to the smallest RMSE is taken as the result.

## Local error correction (LEC)

Since the disturbances affecting $PM_{2.5}$ concentrations are too large to cause sudden changes in monitoring data, this article introduces an error correction method, LEC, to predict the sequence of error values generated by the prediction to realize the correction of the initial prediction values.

Define that there exist two moments before and after t moments and $t − 1$ moments, and the absolute value of the corresponding $PM_{2.5}$ seeking difference is recorded as $\gamma$ as Eq. (3).

$$\gamma = |x(t) − x(t − 1)| \tag{3}$$

where $x(t)$ represents the true value of $PM_{2.5}$ at time t, and $x(t − 1)$ represents the true value of $PM_{2.5}$ concentration at time t − 1. When $\gamma$ satisfies $\gamma \geq a$, the state is called the local mutation state, and the $PM_{2.5}$ concentration point at time t is called the local mutation point. If the $PM_{2.5}$ point corresponding to moment t is the local mutation point, the correction is made as Eq. (4).

$$x\_correct(t) = pred(t) + err\_pred(t) \tag{4}$$

where $x\_correct(t)$ represents the corrected $PM_{2.5}$ prediction at time t; $pred(t)$ represents the initial prediction at time t; $err\_pred(t)$ represents the error prediction at time t. The error series err(t) is derived from the prediction by the CEEMDAN-RLMD-BiLSTM model.

# EXPERIMENT

## Data description

As the capital of China, real-time monitoring and forecasting of local $PM_{2.5}$ is crucial to people's health. In this article, hourly $PM_{2.5}$ data recorded at Station 1010A in Beijing from January 1, 2020, to May 3, 2022, are collated for empirical analysis. Missing data are filled in by the interpolation method.

## Evaluation indicators

In this article, the mean absolute error (MAE), root mean square error (RMSE), and symmetric mean absolute percentage error (SMAPE) are selected as model evaluation indicators.

## Model comparison

To evaluate the performance of the models proposed in this article, several models were selected for comparison.

(1) Traditional machine learning algorithmic models: support vector regression models (SVR).

(2) Neural network models: reverse-pass neural network (BPNN), recurrent neural network (RNN), long and short-term memory neural network (LSTM), gated recurrent unit network (GRU), bidirectional long and short-term memory neural network (BiLSTM), transformer and convolutional neural network (CNN).
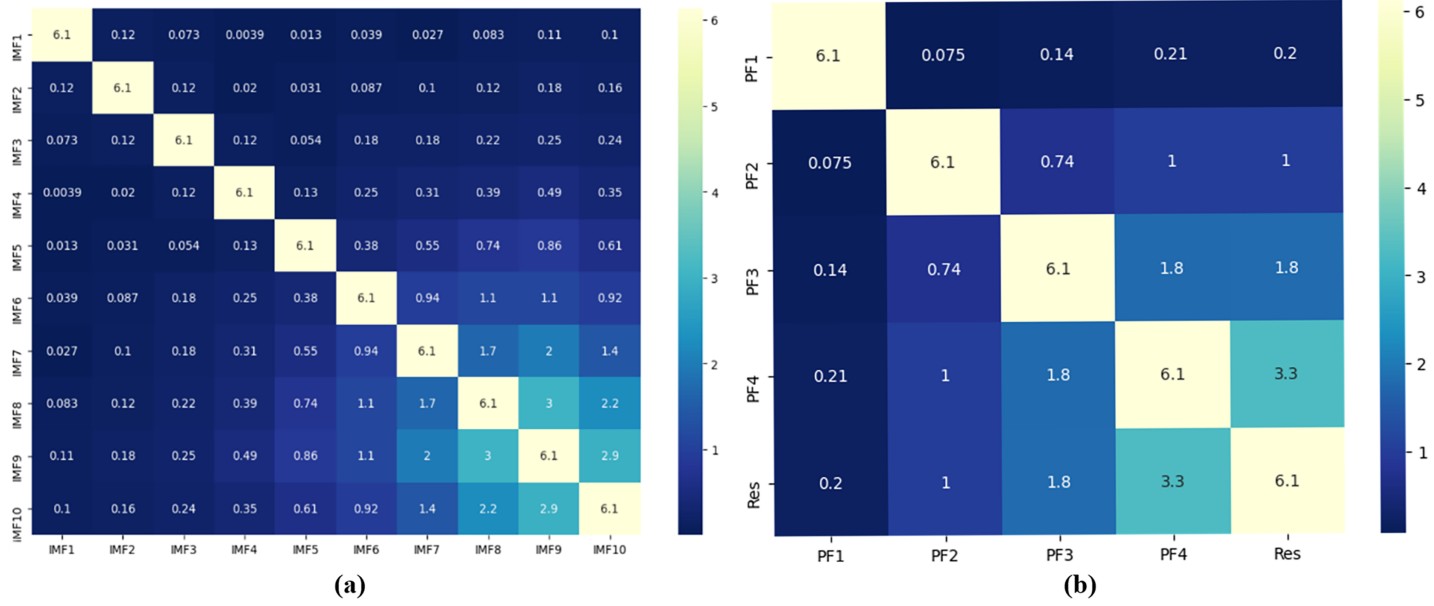

**Figure 2 (A and B) Component correlation (experimental section).** 

(3) Combined models: CEEMDAN-BiLSTM, RLMD-BiLSTM, CEEMDAN-RLMD-BiLSTM.

Due to the sensitivity and uncertainty of $PM_{2.5}$ concentration changes, the input steps of each of our models are divided into 3, 6, 9, and 12, *i.e.*, the first 3, 6, 9, and 12 h of observed data are entered as the predicted values for the next moment. The selection of hyperparameters in the prediction model and the decomposition technique primarily relies on a trial-and-error method. The mean squared error (MSE) serves as the loss function in the BiLSTM neural network, and the weights are optimized using the adaptive momentum estimation method. Implementation of the algorithm is based on MATLAB and Python.

## Decomposition results

Based on the complexity of $PM_{2.5}$ concentration characteristics and the nonstationary of the series, proper decomposition of the raw data is significant for subsequent prediction. First, the original series is decomposed into several fluctuating components with different frequency patterns $IMF_i$ and several product functions $PF_i$ and a residual component using two decomposition methods: CEEMDAN and RLMD. The two are then compared, as shown in Fig. 2.

In Fig. S1, the IMFs and original $PM_{2.5}$ concentration signals from high frequency to low frequency and residual components are presented in order from top to bottom. CEEMDAN decomposes a total of 10 IMF components, and RLMD decomposes a total of 4 PF components and a residual component. To further analyze the decomposition effects of the two decomposition algorithms, noting that the components show obvious nonlinear characteristics, this article utilizes copula entropy to calculate the correlation coefficients between different components and the original sequence two by two, as shown in Fig. 2.

It is worth noting that the full-order correlation derived by CE has better performance in mining the correlation characteristics between data compared to the second-order correlation derived by Pearson and Spearman coefficients (*Ma, 2019*). The backward components of the CEEMDAN and RLMD decompositions are more correlated with the original series compared to the forward components, indicating that the forward components are mainly noisy in the reconstruction part. In the CEEMDAN decomposition, the correlations between $IMF_{10}$ and $IMF_9$, $IMF_{10}$ and $IMF_8$, and $IMF_9$ and $IMF_8$ are 2.9, 2.2, and 3, respectively. Then, it is considered that there is some positive correlation, indicating the existence of component redundancy, which reflects the problem that CEEMDAN decomposition is subject to spurious components. The correlation between the PF4 component and residual in the RLMD decomposition is found to be 3.3, and the remainder has no obvious correlation, reflecting the advantage of RLMD in effectively avoiding mode redundancy issues in CEEMDAN.

To further screen out components with significant characteristics for subsequent prediction, this article introduced permutation entropy (PE) (*Bandt & Pompe, 2002*) to screen the features of each component. The results are shown in Fig. 3. It can be intuitively seen that $PF_1$ in $IMF_1$ and RLMD in CEEMDAN are high-frequency components, and the correlation with other components is low, so they can be considered noise components. $IMF_1$ in CEEMDAN and $PF_1$ in RLMD are eliminated, and the threshold is 0.9.

## Prediction results

In this section, we use the BiLSTM neural network to predict the results of the CEEMDAN decomposition (IMFs 2 through 10) and the RLMD decomposition (PFs 1 through 4 and the residual component) from the previous section, based on a split of the training set (comprising 70%) and the test set (making up the remaining 30%). The results from each component are amalgamated to derive the predicted values from both decomposition algorithms. It is important to underscore that the training phase does not involve any data from the test set. This approach ensures that the decomposition is performed solely on the data from the training set and that the prediction model is trained on subsequences derived from the training set, thereby circumventing any potential leakage of test set data during training. Following several rounds of hyperparameter tuning experiments, we identified the optimal parameter settings and proceeded with the prediction.

To fully evaluate the performance of the model proposed in this article, multiple models mentioned above will be used for comparison, including SVR, BPNN, RNN, LSTM, GRU, BiLSTM, transformer, CNN, CEEMDAN-BiLSTM, RLMD-BiLSTM, and CEEMDAN-RLMD-BiLSTM.

### *Prediction accuracy analysis*

The parameter settings of each model and the prediction results of each model are shown in Table 1 and Table S1.

In all experiments conducted, the proposed CEEMDAN-RLMD-BiLSTM-LEC model showed the smallest MAE, RMSE, and SMAPE, outperforming all other models in these evaluation metrics. These results underscore the superior predictive accuracy of the
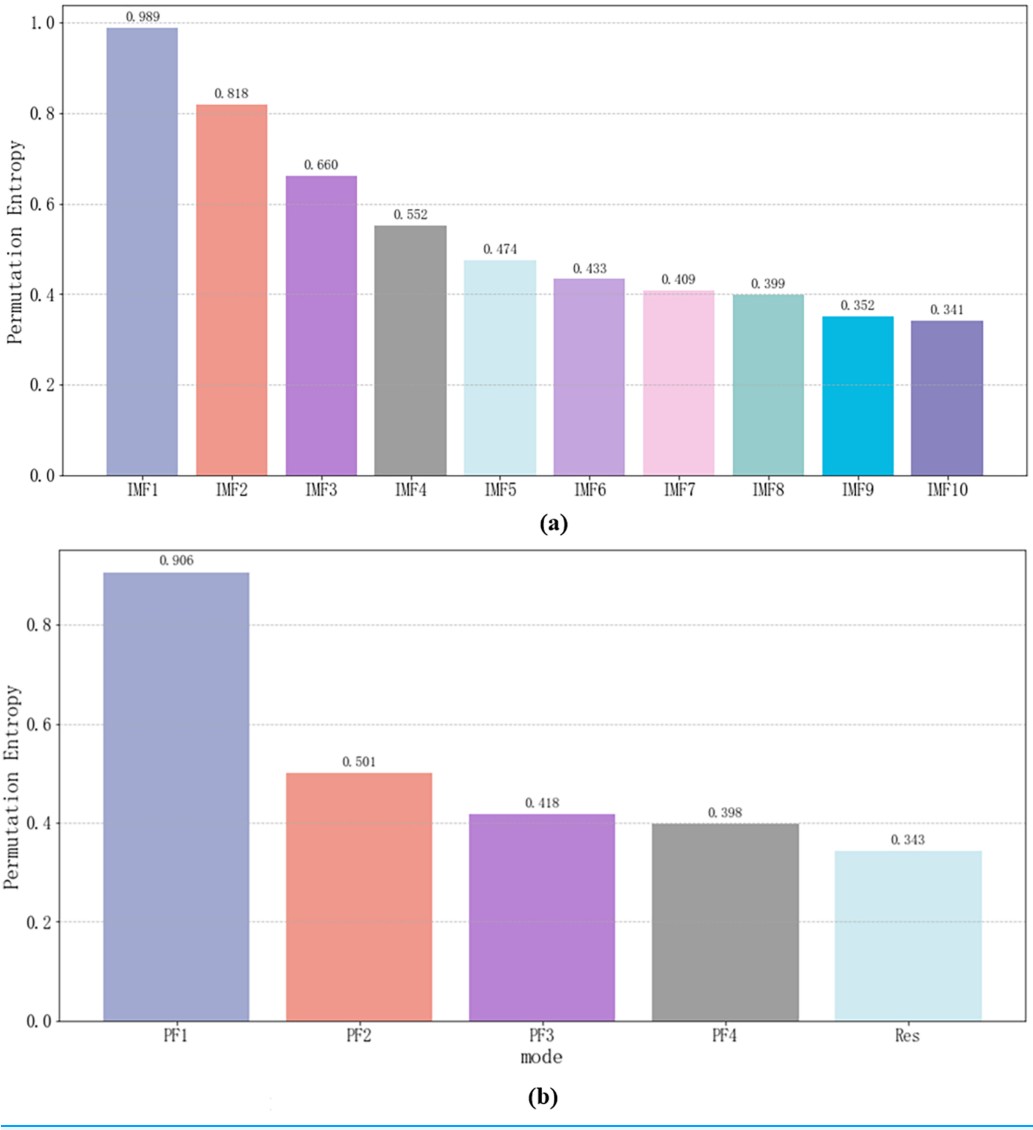

**Figure 3** (A and B) Permutation entropy of each component for the original sequence (experimental section).

CEEMDAN-RLMD-BiLSTM-LEC model in comparison to other comparable models. Among the four input step lengths, the smallest RMSE and SMAPE were both at a step length of 9, while the smallest MAE was at a step length of 12.

For instance, at a 12-step length, the CEEMDAN-RLMD-BiLSTM-LEC model's MAE of 3.9147 μg/m$^3$ was significantly lower than the MAEs of the SVR, BPNN, RNN, LSTM, GRU, BiLSTM, and CNN models, which were 24.1578, 20.1145, 18.3365, 16.4255, 23.1458, 9.8019, and 10.0212 μg/m$^3$, respectively. The MAE and SMAPE metrics also followed a similar trend. Interestingly, the GRU model performed noticeably poorer than the LSTM model, which in turn was outperformed by the BiLSTM model. This is because the GRU model is a simplified version of the LSTM model and might perform weaker with sufficient sample data. The BiLSTM model, with an additional backward LSTM layer, capitalizes on

**Table 1 Parameter settings.**

| Models | Parameters | Settings |
|---|---|---|
| BPNN | Network architecture | $3 \times 100 \times 100 \times 1$ |
| RNN | Hidden units | 200 |
| LSTM | Hidden units | 200 |
| SVR | Kernel function | Radial basis function |
|  | Penalty coefficient | 1 |
| GRU | Hidden units | 200 |
|  | Learning rate | 0.01 |
|  | Regularization parameter | 0.001 |
| BiLSTM | Hidden units | 200 |
|  | Learning rate | 0.01 |
|  | Regularization parameter | 0.001 |
| CNN | Learning rate | 0.01 |
|  | Maxepoch | 500 |

the advantage of bidirectional transfer. However, the CNN model did not perform as well as the BiLSTM model in prediction, and therefore, future decomposition prediction models will no longer involve the BPNN, RNN, LSTM, GRU and CNN models. Additionally, the performance of SVR in time series prediction was considerably weaker than the above neural network models; hence, it will not be discussed further.

Second, when compared to individual decomposition models, the combined decomposition models demonstrated a significant performance boost, showcasing smaller prediction errors and enhanced stability. For instance, at a nine-step length, the SMAPE of the CEEMDAN-RLMD-BiLSTM model was reduced by 0.1374 and 0.1354 compared to the CEEMDAN-BiLSTM and RLMD-BiLSTM models, respectively. Despite RLMD achieving complete decomposition, it lacks comprehensive feature extraction, leading to larger prediction errors. Conversely, although CEEMDAN may have false components, its feature extraction is complete, resulting in smaller prediction errors. This further indicates that the combined prediction models can complement the algorithms, grasp the essence of the original sequence features, and significantly enhance prediction accuracy.

The experimental results indicate that the smallest RMSE for all predictions in the LEC process steps is 9. This suggests that when the difference in $PM_{2.5}$ concentration at adjacent time points exceeds 9, the predictive accuracy of CEEMDAN-RLMD-BiLSTM could be affected, necessitating LEC treatment. For instance, at a 12-step length, the RMSE was reduced by 5.6378% following LEC treatment, with the MAE and SMAPE reduced by 20.5040% and 54.5537%, respectively. This shows that while LEC did not significantly enhance the overall stability of the model's prediction (RMSE), it did markedly improve the model's predictive stability at points of drastic data change (MAE, SMAPE) and reduced errors caused by random fluctuations in $PM_{2.5}$ concentration. Compared to the Transformer, it also outperforms in all metrics, particularly the MAE, highlighting the effectiveness of LEC.
Taking the prediction test set curves of the nine-step and 12-step lengths as representative examples, illustrated in Fig. S2, the performance of each model can be compared against the original sequence. Notably, the CEEMDAN-RLMD-BiLSTM-LEC model demonstrates superior adaptability, maintaining a high degree of fit even in the face of significant local data oscillations. Furthermore, when looking at the overall fit with the entire sequence, this model clearly surpasses the others, providing compelling empirical evidence of its superior predictive capabilities. The robust performance of the CEEMDAN-RLMD-BiLSTM-LEC model underscores its potential as a powerful tool for time-series prediction tasks, particularly in dealing with complex and volatile datasets such as $PM_{2.5}$ concentrations.

### Prediction error analysis

This section delves further into the analysis of prediction errors. Figures 4A and 5A illustrate the actual prediction errors of various comparative models and the proposed model at different time points, using the nine-step and 12-step lengths as examples. The results indicate that the prediction error curve of the CEEMDAN-RLMD-BiLSTM-LEC model is closely aligned with the x-axis, suggesting that at each time point, the prediction errors are approximately distributed around zero, significantly smaller than the prediction errors of other models at corresponding time points. Figures 4B, 4C, 5B and 5C display the histograms of error frequency distributions and kernel density diagrams for all models considered in this study. The distribution of error values for the target models is concentrated within the 5–10 $\mu g/m^3$ range, with a relatively small variance. Moreover, in both step-length experiments, the frequency of the error probability distribution within the 5–10 $\mu g/m^3$ range exceeds 700, significantly presenting the highest kurtosis. These observations provide more profound validation of the accuracy and robustness of the CEEMDAN-RLMD-BiLSTM-LEC model.

## Generalization analysis

To further ascertain the model's accuracy and robustness across diverse datasets, we will expand our experimental analysis to incorporate $PM_{2.5}$ concentration monitoring data from the Beijing 1001A station. Specifically, we will examine the data collected in nine-step and 12-step intervals from 10:00 on February 12, 2022, to 14:00 on December 6, 2022. This rigorous testing approach allows us to validate the effectiveness of our model under different data conditions and assess its generalizability in real-world air quality monitoring scenarios.

### Decomposition results

As depicted in Fig. S3, the RLMD decomposition results in five PF components and one residual component, whereas the CEEMDAN decomposition yields 14 IMF components. Figure 6 further presents the CE correlation coefficients between the original data and the outcomes produced by these two decomposition methods. The decomposition effects adhere to the same principles as those discussed in "Decomposition Results". Upon closer inspection, it becomes evident that, despite the discrepancy in the quantity of decomposed components, both methods exhibit considerable effectiveness in extracting significant
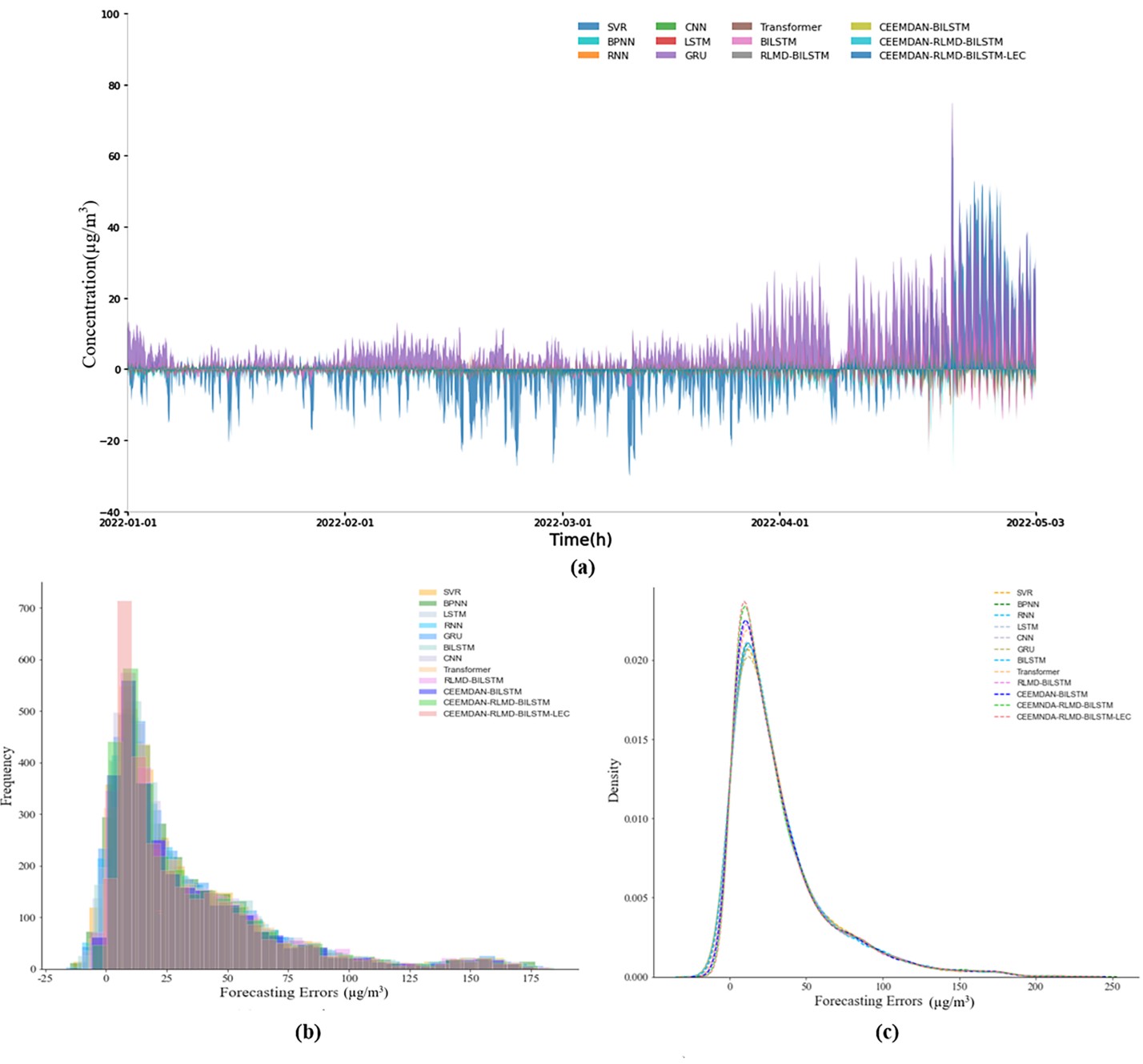

**Figure 4** **(A–C) Prediction errors (nine-step, experimental section).**

features and patterns embedded within the data. This observation provides additional support for our analysis articulated in "Decomposition Results", affirming the high efficiency and precision of both decomposition methods in their designated operations.

Following experimental trials of PE feature selection, it has been discerned that the prediction performance is optimal when RLMD does not remove any component, while for CEEMDAN, the best prediction results are obtained when $IMF_1$ and $IMF_2$ are excluded. Therefore, a PE threshold of 0.95 was chosen (Fig. 7).

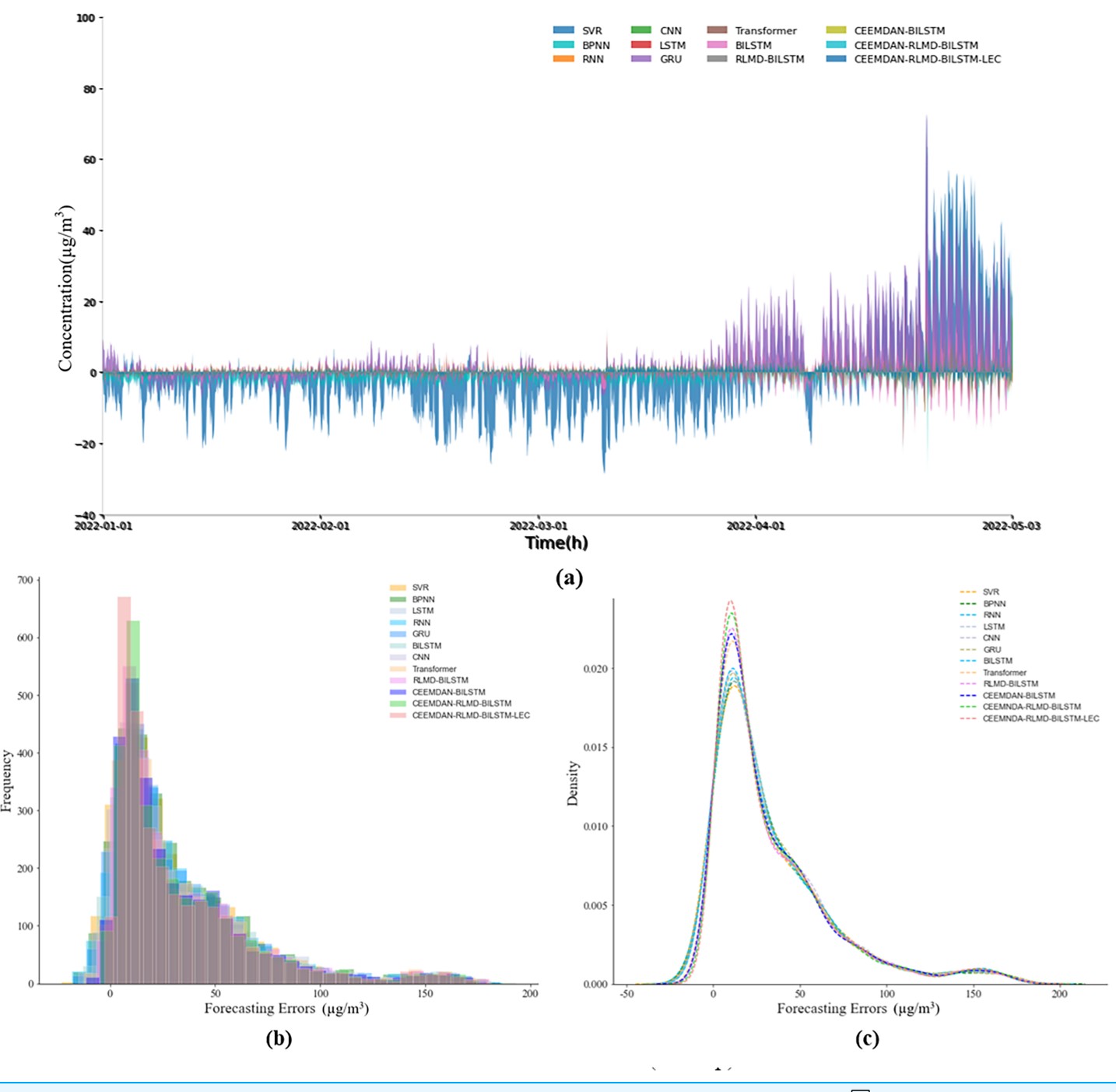

**Figure 5** (A–C) Prediction errors (12-step, experimental section).

### Analysis of prediction results

Building on the conclusions derived from "Prediction Accuracy Analysis", this section will concentrate exclusively on the Transformer, RLMD-BiLSTM, RLMD-BiLSTM, CEEMDAN-RLMD-BiLSTM, and CEEMDAN-RLMD-BiLSTM-LEC models, consequently excluding consideration of the SVR, BPNN, RNN, GRU, LSTM, and CNN models. These selected models have emerged as top performers in our preceding
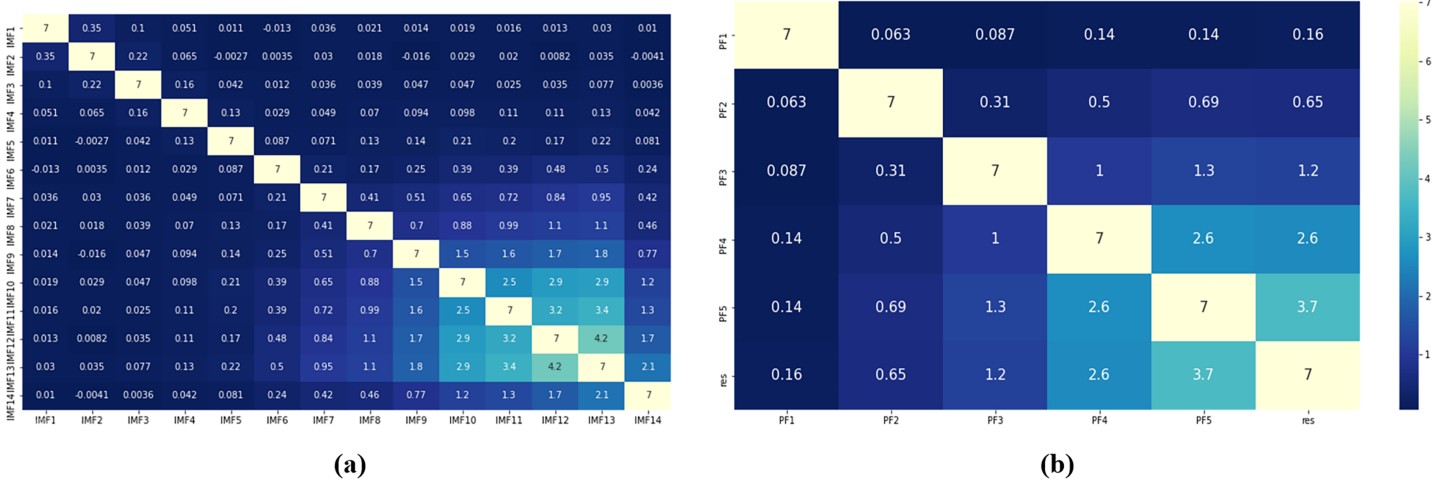

**Figure 6 (A and B) Component correlation (generalization section).**

evaluations. The components they predict comprise IMF$_3$ to IMF$_{14}$, PF$_1$ to PF$_5$, along with the residual component. Our comparative analysis will adhere to the parameter settings detailed in Table 1, thereby ensuring a consistent basis for evaluation. The insights gleaned from this comparative endeavor are presented in Table S2.

The experimental analysis of PM$_{2.5}$ concentration data from Beijing Station 1001A yielded results that aligned with the conclusions presented in "Prediction Accuracy Analysis". For instance, in a nine-step forecast, the RLMD decomposition prediction method produced an MAE of 4.6689 μg/m$^3$, an RMSE of 7.2767 μg/m$^3$, and an SMAPE of 0.1072. These were the least favorable outcomes among the four models scrutinized, reflecting the limitations of RLMD in fully capturing and extracting data features. However, upon integration with CEEMDAN, the values of MAE, RMSE, and SMAPE were reduced to $_{2.5}$320 μg/m$^3$, 3.6558 μg/m$^3$, and 0.0482, respectively. This decrement signifies reductions of 45.7688%, 49.7602%, and 55.0373% compared to the RLMD decomposition prediction, respectively. These results surpassed the performance of the CEEMDAN decomposition prediction by 5.4448%, 3.3981%, and 12.9983%, respectively. These figures illustrate the improved capability of the CEEMDAN-RLMD decomposition prediction in extracting and preserving data features compared to a standalone decomposition prediction. This statement maintains an objective tone devoid of a first-person perspective. Upon the application of LEC, the model findings indicated an LEC threshold of 8, implying that when the absolute difference between two adjacent data points exceeds 8, the incorporation of LEC could reduce the MAE, RMSE, and SMAPE of the model by 27.7291%, 15.2214%, and 19.2946%, respectively. This decrease underscores an enhanced sensitivity of the model to local variations, suggesting that LEC can effectively mitigate random disturbances in PM$_{2.5}$, allowing the model to adapt to significant data shifts and thereby delivering robust predictive performance for data exhibiting high numerical elasticity. The same trend was evident in the 12-step prediction, with the proposed model consistently outperforming the Transformer model.

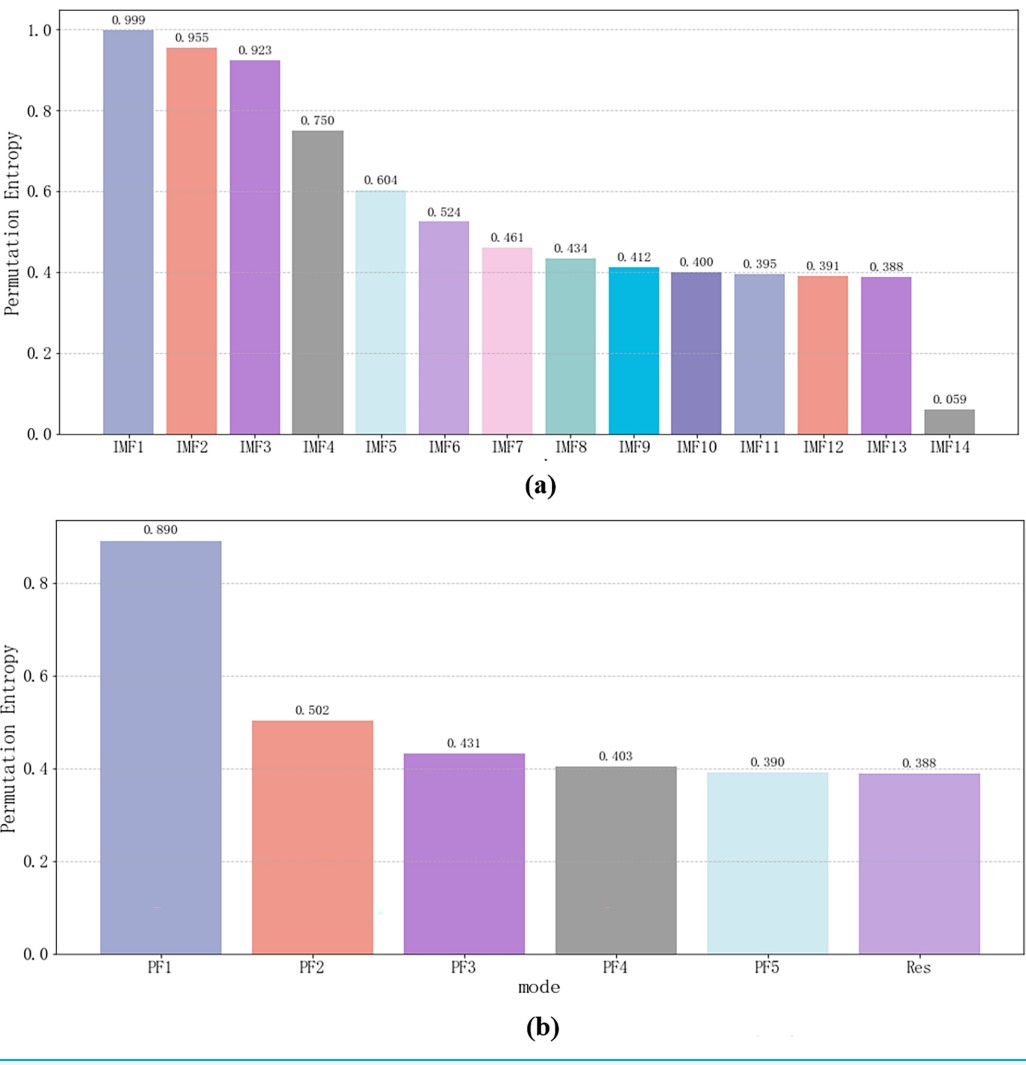

**Figure 7 (A and B) Permutation entropy of each component for the original sequence (generalization section).**

To illustrate the predictive fitting efficiency of the proposed model, Fig. S4 presents the time series curves of the 12 models juxtaposed against the original sequence. The results disclose that CEEMDAN-RLMD-BiLSTM-LEC tightly overlays the original sequence curve. In the face of larger local data fluctuations, it showcases a stronger fit with the original data compared to the other predictive curves, implying that the model is proficient in effectively capturing all the features of the original sequence.

### Prediction error analysis

The analysis in this section further corroborates the outstanding performance of our proposed CEEMDAN-RLMD-BiLSTM-LEC model in the context of generalized experiments. Specifically, Figs. 8 and 9 present the prediction errors at each time point for nine-step and 12-step predictions respectively, along with the corresponding histograms of frequency distributions and kernel density curves. Notably, from Figs. 8A and 9A, it is

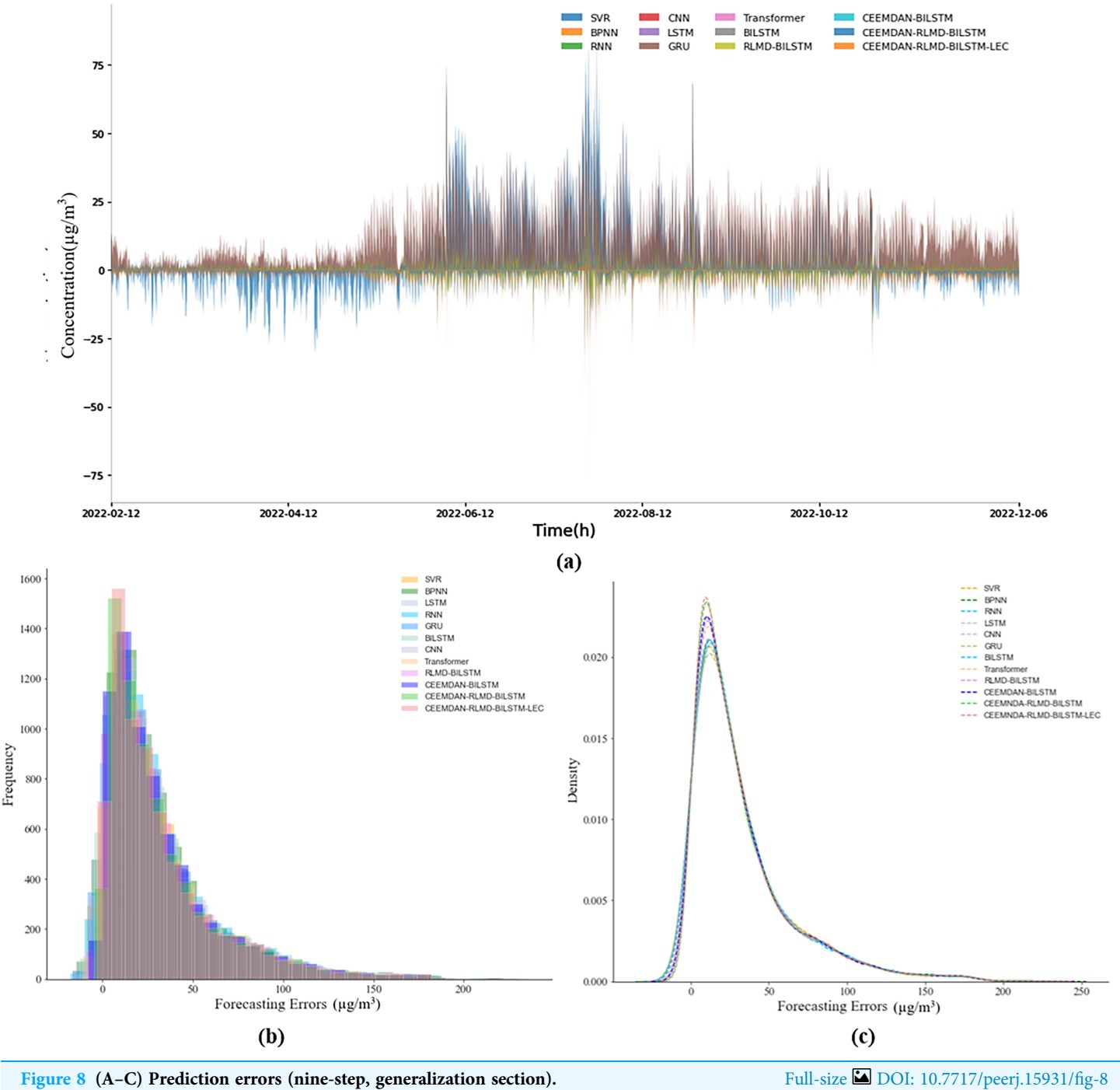

**Figure 8 (A–C) Prediction errors (nine-step, generalization section).**

evident that the prediction errors of the CEEMDAN-RLMD-BiLSTM-LEC model are more concentrated around zero, significantly surpassing other models.

This observation is powerfully reinforced by the frequency distribution curves and kernel density curves depicted in Figs. 8B, 8C, 9B and 9C. Compared to other models, our model exhibits a marked peak in the range of small errors, indicating that its errors conform to a normal distribution with a lower variance. This representation not only

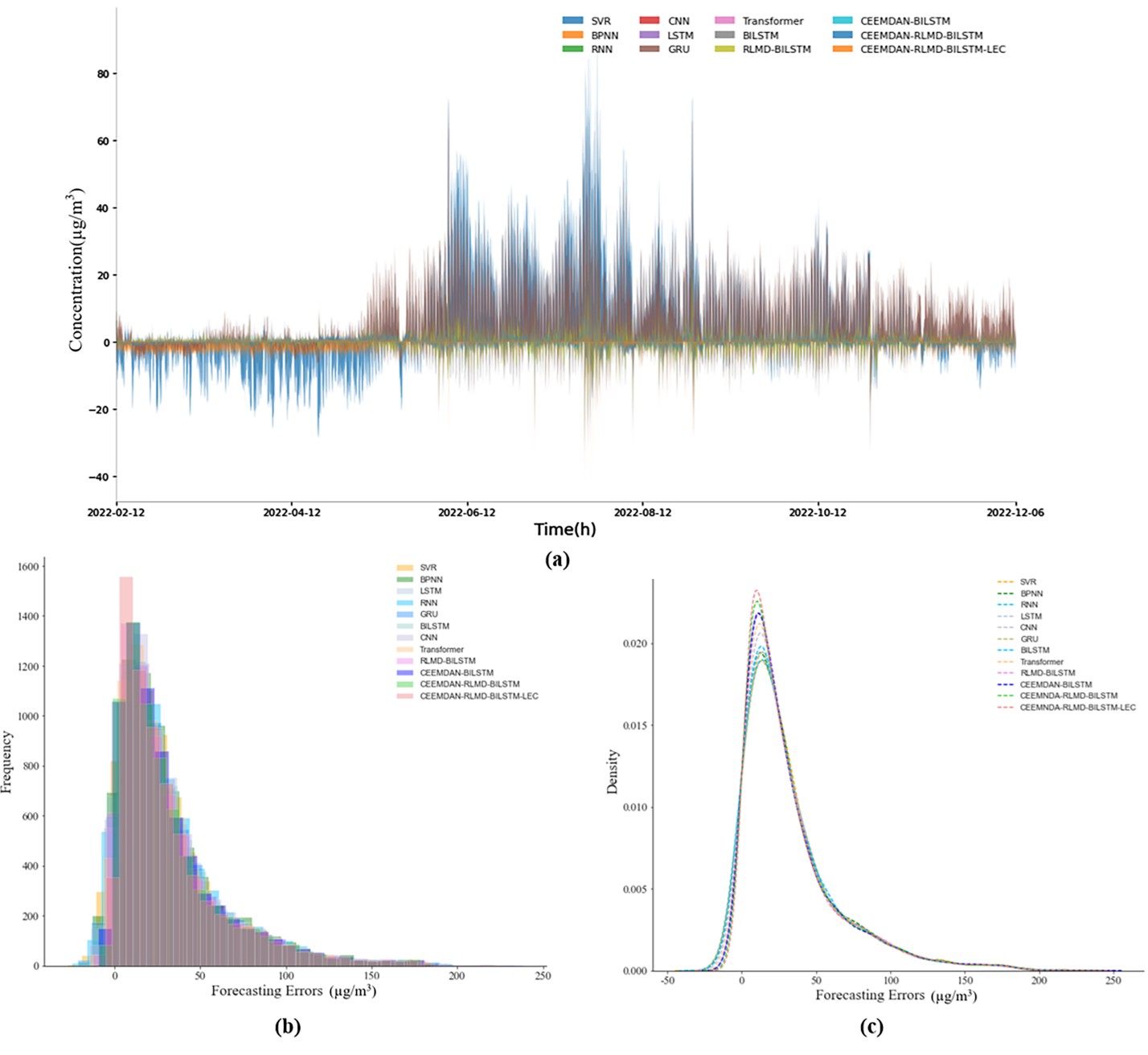

**Figure 9 (A–C) Prediction errors (12-step, generalization section).**

highlights its superiority statistically but also empirically demonstrates that the absolute values of prediction errors for the CEEMDAN-RLMD-BiLSTM-LEC model are relatively small.

Taken together, in combination with the results from "Decomposition Results" and "Analysis of Prediction Results", it can be concluded that the CEEMDAN-RLMD-BiLSTM-LEC model possesses exceptional precision and stability. When dealing with data incorporating noise and disturbances, it demonstrates commendable robustness and

generalization ability. Therefore, we can confidently infer that this model holds considerable application value and broad prospects in tackling various complex and dynamic real-world problems.

## CONCLUSION

This article proposed a $PM_{2.5}$ concentration prediction method that integrates a hybrid decomposition algorithm and a deep learning algorithm, aiming to significantly enhance the accuracy, and robustness of predictions. This method addresses critical challenges in monitoring and managing air quality. Initially, we decompose the $PM_{2.5}$ concentration sequence through CEEMDAN and RLMD. Despite these individual limitations, the strengths of the two methods are complementary. Subsequently, the components from both decomposition algorithms are screened using PE, and the CEEMDAN-RLMD-BiLSTM model is utilized for recombination prediction. Finally, the error prediction set is obtained by comparing the prediction set with the original sequence, and the LEC is applied to handle time series prediction points exceeding the mutation threshold, resulting in the final prediction sequence. Experimental comparisons demonstrate that the combination of RLMD with CEEMDAN significantly enhances the predictive performance compared to individual decomposition methods. The CEEMDAN-RLMD algorithm exhibits smaller errors and demonstrates strong adaptability to sensitive data, confirming its broad applicability.

Therefore, this method improves predictive efficiency and accuracy while maintaining a lower computational complexity so that the proposed model can also be applied to forecast nonstationary and nonlinear time series such as wind power and so on. Furthermore, there are two considerations for improvement: one is to incorporate a temporal convolutional network (TCN) layer to utilize a sufficiently large receptive field to capture more sequence features for deep learning; and the other is to introduce additional influencing variables (such as meteorological factors), along with physics-driven models, for a multi-view analysis.

### Funding
The authors received no funding for this work.

### Competing Interests
The authors declare that they have no competing interests.

### Author Contributions
- Qiao Guo conceived and designed the experiments, performed the experiments, analyzed the data, prepared figures and/or tables, authored or reviewed drafts of the article, and approved the final draft.
- Haoyu Zhang conceived and designed the experiments, performed the experiments, analyzed the data, prepared figures and/or tables, authored or reviewed drafts of the article, and approved the final draft.

- Yuhao Zhang conceived and designed the experiments, performed the experiments, analyzed the data, prepared figures and/or tables, authored or reviewed drafts of the article, and approved the final draft.
- Xuchu Jiang conceived and designed the experiments, performed the experiments, analyzed the data, prepared figures and/or tables, authored or reviewed drafts of the article, and approved the final draft.

## Data Availability

The data is available at Kaggle: https://www.kaggle.com/datasets/xizhouzhang/pm25-concentration-data.

The code is available at GitHub and Zenodo:

- https://github.com/qwaeqa/Prediction-of-PM2.5-concentration-based-on-the-CEEMDAN-RLMD-BILSTM-LEC-model.

- Haoyu Zhang. (2023). qwaeqa/Prediction-of-PM2.5-concentration-based-on-the-CEEMDAN-RLMD-BILSTM-LEC-model: v1.0.0 (v1.0.0). Zenodo. https://doi.org/10.5281/zenodo.8140823.

## Supplemental Information

Supplemental information for this article can be found online at http://dx.doi.org/10.7717/peerj.15931#supplemental-information.

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
