# Peer review of "Prediction of PM2.5 concentration based on the CEEMDAN-RLMD-BiLSTM-LEC model"

_PeerJ, doi:10.7717/peerj.15931_

## Round 0.1 · original submission · Major Revisions

Based on the referees' comments on your manuscript, major revisions are required to further improve the quality.

Reviewer 1 ·

Basic reporting

The article is clear and unambiguous, and prior literature is appropriately referenced. The structure of the article conforms to an acceptable format.

Experimental design

This paper aims at proving the effectiveness of the combination of the decomposition model and the machine learning model. This kind of combination is endless. For instance, they can ensemble CNN or Transformer with other decomposition models. Why do authors not try these combinations? It can not be proved that the experimental design in the article is reasonable.

Validity of the findings

no comment

Reviewer 2 ·

Basic reporting

This paper provides a nonparametric regression-based approach to explore the relationship between air pollution and PM concentration. This method can help researchers better understand the temporal and spatial variation of PM concentration, and provide scientific basis for formulating corresponding environmental protection policies. Although the research results are compelling, there are still some suggestions I would like to propose to help improve the quality.
1. Based on the 16 IMF components obtained by CEEMDAN decomposition and the 5 PF components obtained by RLMD decomposes, each component should be screened by their influencing degree on the predicted label, only components with higher degree can be selected as input indicators.
2. It is noted that correlation analysis was conducted on variables obtained by CEEMDAN decomposition and RLMD decomposes respectively, however the reason why some variables get selected as input components, and others not should be further explained.
3. The experiment needs to add more details. I suggest that you improve the description in lines 314- 344 to provide more justification for your study.
4. To test whether the model is still accurate and robust to different data, was the model re-trained when analyzing data from Beijing 1001A station?
5. In this study, to attain fluctuation characteristic, pre-decomposition process is needed for prediction validation on every run. However PM2.5 prediction is real-time prediction, how to maintain prediction accuracy when facing insufficient decompostion characteristic due to lack of contiguous data on time scale
6.It is highly recommended to provide the raw data, however, the supplemental files need more descriptive metadata identifiers to be useful to future readers.

Experimental design

No comments

Validity of the findings

No comments

Additional comments

No comments

Annotated reviews are not available for download in order to protect the identity of reviewers who chose to remain anonymous.

Reviewer 3 ·

Basic reporting

No comment

Experimental design

1) The figures representing the correlation (Figure 4, Figure 9): the numbers are not presented clearly and should not be stretched to make the numbers shrink (Figure 4a, Figure 9a).
2) The presentation of the unit in the figure (ug/m3) (Figures 5, 6, 7, 10, 11, and 12) should follow the format of PeerJ.

Validity of the findings

1. Lines 197–198: 23,956 data points belong to which particular region or monitoring station in India? It is necessary to provide a map to show the locations of all sites used in this study.
2. Also, lines 197–198 correspond to Figure 1: the data from June 4, 2019 to June 4, 2022 is 4 years, not 5 years as shown in Figure 1.
3. In addition to using PM2.5 concentration as a dependent variable in the data for PM2.5 prediction models, independent variables (often meteorological variables) are also required. What independent variables were used in this study? In this case, authors should include more information about this in the data description.
4. Is a prediction for PM2.5 concentrations for a specific future time period available? (Ex: 7-day short-term prediction).

Annotated reviews are not available for download in order to protect the identity of reviewers who chose to remain anonymous.

---

## Round 0.2 · accepted · Accept

Your manuscript can be accepted based on the referee's comment.

Reviewer 2 ·

Basic reporting

The quality of this paper has been substantially enhanced after thorough revision. No extra comments are given.

Experimental design

No comment

Validity of the findings

No comment

Additional comments

No comment